# The impact of reduced skeletal muscle mass at stroke onset on 3-month functional outcomes in acute ischemic stroke patients

**Kyong Young Kim[1], Seunguk Jung[2,3]\*, Eun Bin Cho[2,3], Tae-Won Yang[2,3], Seung Joo Kim[2], Hyunsung Kim[2], Sunhye Jung[2]**

**1** Department of Endocrinology, Gyeongsang National University Changwon Hospital, Changwon, Republic of Korea, **2** Department of Neurology, Gyeongsang National University Changwon Hospital, Changwon, Republic of Korea, **3** Department of Neurology and Institute of Health Science, Gyeongsang National University College of Medicine, Jinju, Republic of Korea

\* seunguk1358@gmail.com

## Abstract

### Introduction

Sarcopenia, characterized by reduced skeletal muscle mass (RMM), is increasingly recognized as a significant factor influencing outcomes in various health conditions, including stroke. Although most studies focus on sarcopenia developing during stroke rehabilitation, the impact of sarcopenia present at the onset of acute ischemic stroke remains underexplored. This study aims to evaluate the effect of RMM at stroke onset on 3-month functional outcomes in acute ischemic stroke patients.

### Materials and methods

We prospectively enrolled acute ischemic stroke patients admitted between May 2019 and December 2019. Muscle mass was accessed early during hospitalization using whole-body dual-energy X-ray absorptiometry (DXA), and patients were categorized into RMM and normal muscle mass (NMM) groups based on the Asian Working Group for Sarcopenia (AWGS) criteria. Functional outcomes at 3 months were assessed using the modified Rankin Scale (mRS), with unfavorable outcomes defined as mRS scores 2–5. Multivariable logistic regression and SHAP (Shapley Additive exPlanations) analyses were used to evaluate the independent impact of RMM on 3-months functional outcomes.

### Results

A total of 99 patients were analyzed. The RMM group had a significantly higher prevalence of unfavorable outcomes at 3 months compared to the NMM group (p < 0.001). Patients with RMM were older and presented with more severe strokes. Multivariable analysis confirmed RMM as an independent predictor of unfavorable outcomes (adjusted OR: 8.07, 95% CI: 1.603–40.66, p = 0.011), even after adjusting for age and initial stroke severity. SHAP analysis ranked RMM as the second most influential predictor of unfavorable outcomes,

**Data Availability Statement:** Due to ethical restrictions, de-identified data cannot be shared as it includes sensitive patient information. These restrictions were imposed by the Institutional

Review Board (IRB approval No. 2019-03-005). Interested researchers can request data access by contacting the IRB directly at https://www.r-bay.co.kr or via telephone at +82-55-214-3701.

**Funding:** This research was supported by the Lee Jung Ja research grant of Gyeongsang National University Hospital (LJJ-GNUH-2019-0001). The funders had no role in study design, data collection and analysis, decision to publish, or preparation of the manuscript.

**Competing interests:** The authors have declared that no competing interests exist.

following NIHSS on admission. These findings indicate that RMM not only worsens initial stroke severity but also independently hinders post-stroke recovery.

## Conclusions

Reduced muscle mass at the onset of acute ischemic stroke is a significant, independent predictor of unfavorable outcomes at 3 months. In addition to its impact on recovery, RMM is linked to older age and more severe strokes, worsening prognosis. Maintaining muscle mass is also crucial for stroke prevention, as it supports cardiovascular health and resilience. Early identification and intervention for sarcopenia can improve recovery and reduce future stroke risk.

## Introduction

Stroke is a leading cause of disability and mortality worldwide, significantly affecting patients' quality of life and posing a substantial burden on healthcare systems [1]. Functional outcomes after stroke are influenced by a variety of factors, including the severity of the initial neurological deficit, comorbidities, and the patient's overall health status [2]. Sarcopenia, defined as the progressive loss of skeletal muscle mass and strength, is a common condition among older adults and those with chronic diseases [3]. It has been increasingly recognized as a critical factor that adversely affects the prognosis of various conditions, including cardiovascular diseases [4].

In acute ischemic stroke, sarcopenia is increasingly recognized as a factor influencing recovery. However, most studies focus on sarcopenia that develops during stroke rehabilitation, leaving the impact of sarcopenia present at stroke onset underexplored [5]. Recent studies have shown that low appendicular skeletal muscle mass (ASM) measured early during hospitalization in acute stroke patients is associated with poorer functional outcomes at 3 months. Multivariate analysis confirmed that low ASM independently predicted unfavorable outcomes, emphasizing the value of early sarcopenia detection [6]. A systematic review and meta-analysis found that sarcopenia significantly increases the risk of poor physical function in stroke patients, emphasizing the need for early identification and management of sarcopenia in clinical practice [7].

However, despite the growing recognition of sarcopenia's impact on stroke outcomes, its effect on clinical outcomes, particularly during acute phase of stroke, remains unclear. Previous studies have reported conflicting results regarding the relationship between sarcopenia and stroke severity, and the precise mechanisms through which sarcopenia influences stroke recovery remain inadequately explored. This study aims to evaluate the impact of reduced muscle mass at stroke onset on the 3-month functional outcomes in acute ischemic stroke patients.

## Materials and methods

### Study population

We screened consecutive patients with acute ischemic stroke within 7 days of onset at our hospital from May 2019 to December 2019 (N = 282). We included patients aged 18 years or older with acute ischemic stroke confirmed by computed tomography (CT) or magnetic resonance imaging (MRI), presenting within 7 days of onset, with premorbid mRS scores below 2, and

without conditions significantly affecting skeletal muscle mass. All participants provided informed consent. The exclusion criteria were as follows: patients with premorbid modified Rankin Scale (mRS) scores of 2 or more, patients with aphasia, patients with decreased consciousness at the time of stroke onset, patients with advanced dementia, and patients with catabolic conditions including cancer, chronic kidney disease (stage 4 or higher), congestive heart failure, and liver cirrhosis. Additionally, patients who did not consent to participate in the study were also excluded. The reasons for non-participation were unknown for 9 individuals. In total, 99 patients voluntarily provided informed consent and were included in the final analysis. (S1 Fig) Clinical and radiologic data were reviewed. Clinical data included demographic information, time from the onset of symptoms (or last known normal) to arrival at the emergency department (Onset-to-Door time), baseline National Institutes of Health Stroke Scale (NIHSS) and modified Rankin Scale (mRS) scores at discharge and three months after discharge. Baseline laboratory findings, the presence of underlying conditions, and blood pressure at hospital arrival were also recorded. The use of intravenous thrombolysis and intra-arterial therapy at the time of admission was noted, and imaging data, including CT and/or MRI, were analyzed to classify the type of ischemic stroke. This study was prospectively conducted with the approval of our institutional review board (IRB approval No. 2019-03-005). Written informed consent was obtained from all participants (or their legal representatives) prior to their inclusion in the study. All personal data were anonymized to ensure participant privacy and confidentiality.

## Muscle mass measurement

In this study, appendicular skeletal muscle mass (ASM) was measured using whole-body dual-energy X-ray absorptiometry (DXA) with a Hologic DXA system (Hologic Inc., Marlborough, MA, USA). DXA scans were performed early during hospitalization to minimize the impact of acute stroke on muscle mass measurements and to capture skeletal muscle mass shortly after stroke onset. The ASM was calculated by summing the lean mass of the arms and legs, which are the primary sites reflecting overall body muscle status. To determine the skeletal muscle index (SMI), the height-adjusted ASM was calculated using the formula: $SMI = ASM(kg)/height(m)^2$. Patients were then categorized into reduced muscle mass (RMM) and normal muscle mass (NMM) groups based on the Asian Working Group for Sarcopenia (AWGS) criteria, with cut-off values of $<7.0 \ kg/m^2$ for men and $<5.4 \ kg/m^2$ for women [8]. Although muscle power was assessed using handgrip strength using a Takei T.K.K. 5401 GRIP-D handgrip dynamometer (Takei Scientific Instruments Co., Ltd., Niigata, Japan), we found that acute stroke patients often faced challenges due to cognitive impairment, poor cooperation, weakness, and ataxia. For instance, the presence of weakness in the dominant hand necessitates the measurement of hand grip strength using the non-dominant hand. It can reduce the reliability of the assessment. These limitations rendered hand grip strength measurements less reliable in this population. Therefore, we relied on SMI, a relatively objective measure, to classify patients into the groups for analysis in this study. Additionally, we analyzed outcome differences not only between groups defined by the presence or absence of sarcopenia according to the AWGS criteria, including both SMI and handgrip strength results, but also within sex-specific ASM tertile groups.

## Outcome measurement

Outcome measurement in this study was centered on clinical evaluation conducted three months after the onset of stroke. Functional disability was assessed using the mRS scores, a widely accepted and validated tool for measuring the degree of disability or dependence in

daily activities among stroke survivors. The mRS scores ranges from 0 to 6, where a score of 0 indicates no symptoms at all, 1 represents no significant disability despite some symptoms, and higher scores represent increasing levels of disability, with a score of 6 indicating death. For the purpose of analysis, we dichotomized the mRS scores to create two distinct outcome groups based on functional dependency at the 3-month follow-up. Patients with a mRS score of 1 or less were classified into the "favorable outcome" group. This group included patients who were functionally independent, with minimal or no disability, and who were able to perform activities of daily living without assistance. On the other hand, patients with a mRS score of greater than 1 were categorized into the "unfavorable outcome" group. These patients exhibited varying degrees of functional dependency, ranging from requiring some assistance with daily activities to severe disability or death.

## Statistical tests

Continuous variables were presented as mean ± standard deviation (SD) or median with interquartile range (IQR) based on their distribution, and categorical variables were expressed as frequencies and percentages. The normality of continuous data was assessed using the Shapiro-Wilk test. Comparisons between RMM and NMM groups were conducted using the independent t-test or Mann-Whitney U test for continuous variables, depending on the normality of the data, and the chi-square test or Fisher's exact test for categorical variables. For outcome analysis, logistic regression models were employed to assess the association between the groups and functional independence at 3 months. Covariates included in the regression models were age, sex, baseline NIHSS score, presence of comorbidities and laboratory findings to control for potential confounders. The impact of RMM on outcomes was further evaluated using multivariable regression analysis to adjust for significant baseline differences. Odds ratios (ORs) with 95% confidence intervals (CIs) were calculated to quantify the association between low SMI and poor outcomes. Statistical significance was set at a p-value $< 0.05$. All tests were two-tailed, and results were interpreted in the context of clinical relevance. All statistical analyses were performed using Stata/SE 13.0 (StataCorp, College Station, TX, USA). We performed independent variable importance analysis using SHAP (SHapley Additive exPlanations) values to determine the relative contribution of each predictor to the 3-month functional outcomes. SHAP values were computed using a logistic regression model with overlap weighting to enhance covariate balance. The importance of each variable was visualized using SHAP summary plots and variable importance plots, providing insights into the direction and magnitude of their effects

## Results

The analysis revealed that RMM was significantly associated with poor functional outcomes, with a higher prevalence observed in the unfavorable outcome group (93.3%) compared to those with a favorable outcome group (53.6%) ($p < 0.001$). Additionally, patients in the unfavorable outcome group were older (mean age 68.9 years vs. 62.1 years, $p = 0.007$) and had a higher prevalence of sarcopenia (42.86% vs. 21.21%, $p = 0.032$). The lowest ASM tertile (Q1) was also more common in the unfavorable outcome group (53.33% vs. 24.64%, $p = 0.012$). A higher initial NIHSS score on admission was noted in the unfavorable outcome group (median 5 [IQR 3–8] vs. 2 [IQR 1–4], $p < 0.001$), and a history of previous TIA or stroke was more prevalent in this group (30.00% vs. 10.14%, $p = 0.014$). The large artery atherosclerosis (LAA) subtype was also more frequent in the unfavorable outcome group (43.33% vs. 29.41%, $p = 0.046$). These findings indicate that RMM and related factors, including age and higher

NIHSS scores, are important predictors of worse functional outcomes in acute ischemic stroke patients. (Table 1)

Multivariable analysis incorporating factors found to be significant in univariate analysis revealed several independent predictors of poor outcomes in acute ischemic stroke patients. RMM was identified as a significant predictor, with an adjusted odds ratio (aOR) of 8.07 (95% CI: 1.603–40.66, p = 0.011), indicating a strong impact on prognosis. Admission NIHSS score also remained an independent predictor of outcome, with an aOR of 0.84 (95% CI: 0.72–0.98, p = 0.026). Additionally, patients classified under the 'Others' category of the TOAST classification had significantly worse outcomes, with an aOR of 14.89 (95% CI: 1.35–164.31, p = 0.028). These findings highlight RMM, baseline NIHSS, and specific TOAST classifications as key independent factors influencing prognosis in acute ischemic stroke patients. (Table 2)

Given the strong association of RMM with poor outcomes, this relationship may partly reflect its connection to other prognostic factors. To investigate this further, we conducted an independent variable importance analysis using SHAP (SHapley Additive exPlanations) values. This method quantified RMM's unique contribution to outcome prediction, independent of its correlation with other variables. SHAP analysis identified RMM as the second most influential predictor of unfavorable outcomes, following the NIHSS score on admission. SHAP summary and variable importance plots (Fig 1) visually confirmed these findings, highlighting RMM's substantial independent impact. The SHAP summary plot (Fig 1B) revealed that NIHSS on admission contributed most to poor outcomes, with higher scores consistently linked to unfavorable results. RMM, the second-ranked variable, showed positive SHAP values, emphasizing its robust independent association with poor prognosis. Age was the third most significant predictor, with older age contributing to worse outcomes. LDL cholesterol and BMI demonstrated moderate influence with non-linear effects, reflecting the complexity of their roles in recovery. Other factors, such as Onset-to-Door time and diabetes mellitus, had lower SHAP values, indicating a more limited but context-dependent impact. The variable importance plot (Fig 1A) reinforced these findings, with NIHSS on admission and RMM showing the highest mean absolute SHAP values.

The comparison of baseline characteristics between patients with RMM and NMM revealed several significant differences. Patients with RMM were older (mean age 66.4 years vs. 59.97 years, p = 0.009) and had a lower BMI (23.4 vs. 26.7, p < 0.001) compared to those with NMM. The NIHSS score on admission was higher in the RMM group (median 3 [IQR 1–6] vs. 2 [IQR 0–5], p = 0.015), indicating more severe initial neurological impairment. Additionally, diastolic blood pressure was significantly lower in the RMM group (87.6 mmHg vs. 95.1 mmHg, p = 0.025). A history of previous TIA or stroke was more prevalent in the RMM group (21.5% vs. 5.9%, p = 0.049). Laboratory findings showed that ALT levels were lower (p = 0.044) and INR was higher (p = 0.032) in the RMM group. These differences suggest that patients with RMM may present with distinct clinical and laboratory profiles that could impact their stroke outcomes. (S1 Table)

## Discussion

This study investigated the impact of RMM at the onset of acute ischemic stroke on 3-month functional outcomes. Our findings demonstrate that RMM, as measured by appendicular skeletal muscle mass (ASM), is significantly associated with unfavorable functional outcomes, defined as 2–5 mRS scores at 90 days. These results are consistent with previous studies that have identified sarcopenia as a critical determinant of prognosis in various chronic conditions, including cardiovascular disease, cancer and stroke [4, 6, 7, 9, 10].

**Table 1. Comparison of clinical and laboratory characteristics between favorable and unfavorable outcome groups in acute stroke patients.**

| | Favorable outcome group (n = 65) | Unfavorable outcome group (n = 34) | *p*-value |
|---|---|---|---|
| Age, mean (SD), years | 62.1 (11.8) | 68.9 (10.04) | 0.007 |
| BMI, mean (SD), kg/m$^2$ | 24.7 (3.25) | 24.2 (3.09) | 0.519 |
| Reduced muscle mass, n (%) | 37 (53.6) | 28 (93.3) | <0.001 |
| Sarcopenia, n (%) | 14 (21.21) | 12 (42.86) | 0.032 |
| ASM_tertile, n (%) | | | 0.012 |
| Q1 | 17 (24.64) | 16 (53.33) | |
| Q2 | 24 (34.78) | 9 (30.0) | |
| Q3 | 28 (40.58) | 5 (16.67) | |
| Male gender, n (%) | 51 (73.9) | 24 (80.0) | 0.516 |
| NIHSS on admission, median [IQR] | 2 [1–4] | 5 [3–8] | < 0.001 |
| Onset-to-Door time, median [IQR], hours | 5.63 [2.08–13.42] | 6.06 [2.5–13.82] | 0.507 |
| Recanalization therapy, n (%) | | | 0.564 |
| None | 54 (78.26) | 25 (83.33) | |
| IV tPA only | 7 (10.14) | 1 (3.33) | |
| IAT only | 4 (5.80) | 3 (10.0) | |
| combined | 4 (5.80) | 1 (3.33) | |
| SBP, initial, mean (SD), mmH$_2$O | 160 [150–188] | 170 [156–190] | 0.742 |
| DBP, initial, mean (SD), mmH2O | 90 [80–100] | 86.5 [80–90] | 0.136 |
| Risk factors, n (%) | | | |
| HTN | 40 (57.97) | 22 (73.33) | 0.147 |
| DM | 19 (27.54) | 12 (40.00) | 0.219 |
| Dyslipidemia | 38 (55.07) | 16 (53.33) | 0.873 |
| Coronary heart disease | 6 (8.70) | 3 (10.00) | 0.836 |
| Atrial fibrillation | 12 (17.39) | 5 (16.67) | 0.93 |
| Previous TIA or Stroke | 7 (10.14) | 9 (30.00) | 0.014 |
| Smoking within 5 years | 29 (42.03) | 13 (43.33) | 0.904 |
| TOAST classifications, n (%) | | | 0.046 |
| LAA | 20 (29.41) | 13 (43.33) | |
| SVO | 22 (32.35) | 10 (33.33) | |
| CE | 9 (13.24) | 6 (20.00) | |
| Others | 17 (25.0) | 1 (3.33) | |
| Laboratory findings, mean (SD) | | | |
| WBC counts, ×10$^3$/μL | 7.898 (2.35) | 8.64 (3.43) | 0.216 |
| Hemoglobin, g/dL | 13.98 (1.84) | 14.14 (1.93) | 0.687 |
| Hematocrit, % | 40.54 (4.88) | 40.5 (5.18) | 0.974 |
| Platelet counts, ×10$^3$/μL | 231.6 (62.4) | 225.03 (118.3) | 0.719 |
| Total cholesterol, mg/dL | 196.9 (49.6) | 181.9 (40.2) | 0.149 |
| Triglyceride, mg/dL | 144.1 (83.5) | 128.1 (56.6) | 0.342 |
| HDL, mg/dL | 46.3 (11.4) | 43.7 (9.04) | 0.277 |
| LDL, mg/dL | 112.5 (32.2) | 102.4 (28.1) | 0.137 |
| Laboratory findings, median [IQR] | | | |
| Fasting glucose, mg/dL | 130 [108–156] | 116.5 [108–150] | 0.613 |
| HbA1c, % | 6 [5.6–6.4] | 6 [5.5–6.5] | 0.828 |
| BUN, mg/dL | 15.1 [12.4–19.8] | 17.1 [13.9–21.1] | 0.295 |
| Creatinine, mg/dL | 0.82 [0.68–0.99] | 0.79 [0.68–1.04] | 0.869 |
| Cockcrofr-Gault eGFR, mL/min | 91 [80–102] | 90 [65–98] | 0.379 |
| AST, U/L | 23 [20–30] | 25 [22–28] | 0.618 |

*(Continued)*

**Table 1.** (Continued)

| | Favorable outcome group (n = 65) | Unfavorable outcome group (n = 34) | *p*-value |
|---|---|---|---|
| ALT, U/L | 22 [15–30] | 18 [15–27] | 0.419 |
| ESR, mm/h | 9 [4–18] | 11 [5–17] | 0.398 |
| hs CRP, mg/L | 1.1 [0.6–2.6] | 1.45 [0.5–3.15] | 0.579 |
| INR | 0.96 [0.92–1.02] | 0.99 [0.95–1.03] | 0.127 |
| aPTT, seconds | 33.2 [31.1–35.4] | 33.45 [30.8–36.2] | 0.749 |
| Fibrinogen, mg/dL | 322 [287.5–364.5] | 324 [301–371.5] | 0.327 |
| NT-proBNT, pg/mL | 129.5 [39–314] | 72 [41–258] | 0.526 |

SD: Standard Deviation, BMI: Body Mass Index, SMI: Skeletal Muscle Index, ASM: Appendicular Skeletal Muscle, NIHSS: National Institutes of Health Stroke Scale, IQR: Interquartile Range, IV tPA: intravenous tissue plasminogen activator, IAT: intra-arterial therapy, SBP: Systolic Blood Pressure, DBP: Diastolic Blood Pressure, HTN: Hypertension, DM: Diabetes Mellitus, TIA: Transient Ischemic Attack, TOAST: Trial of Org 10172 in Acute Stroke Treatment, LAA: Large Artery Atherosclerosis, SVO: Small Vessel Occlusion, CE: Cardioembolism, WBC: White Blood Cells, HDL: High-Density Lipoprotein, LDL: Low-Density Lipoprotein, BUN: Blood Urea Nitrogen, HbA1c: Hemoglobin A1c, eGFR: Estimated Glomerular Filtration Rate, AST: Aspartate Aminotransferase, ALT: Alanine Aminotransferase, ESR: Erythrocyte Sedimentation Rate, hsCRP: High-Sensitivity C-Reactive Protein, INR: International Normalized Ratio, aPTT: Activated Partial Thromboplastin Time, NT-proBNP: N-terminal pro b-type Natriuretic Peptide, RHI: Reactive Hyperemia Index, LnRHI: Natural Logarithm of the Reactive Hyperemia Index

Several studies have emphasized the negative impact of sarcopenia on recovery after stroke. A systematic review and meta-analysis reported that sarcopenia significantly increases the risk of poor physical function in stroke patients, underscoring the importance of early identification and intervention to mitigate its effects [7]. Our findings align with previous studies that have highlighted the detrimental effects of sarcopenia on recovery in acute ischemic stroke. Our study similarly found that even after adjusting for all factors related to stroke prognosis, such as age, stroke severity at onset, history of previous stroke or transient ischemic attack (TIA), and stroke subtype, RMM remained significantly associated with unfavorable outcomes. The remarkably strong association between reduced muscle mass (RMM) and unfavorable outcomes (OR > 8) compared to other recognized markers, which displayed relatively modest odds ratios, may partly be explained by the close relationship between RMM and other factors influencing poor prognosis including age. Reduced skeletal muscle mass and function is associated with systemic inflammation, metabolic dysfunction,

**Table 2. Multivariate analysis of factors associated with functional outcomes in acute ischemic stroke patients.**

| | OR | 95% CI | *p*-value | aOR | 95% CI | *p*-value |
|---|---|---|---|---|---|---|
| Age, per year | 0.945 | 0.905–0.986 | 0.009 | 0.97 | 0.92–1.02 | 0.187 |
| Reduced muscle mass | 12.108 | 2.673–54.841 | 0.001 | 8.07 | 1.603–40.66 | 0.011 |
| NIHSS on admission, per 1 point | 0.854 | 0.764–0.955 | 0.006 | 0.84 | 0.72–0.98 | 0.026 |
| History of Stroke or TIA | 0.263 | 0.087–0.795 | 0.018 | 0.27 | 0.06–1.19 | 0.083 |
| TOAST classifications | | | | | | |
| LAA | Ref. | | | Ref. | | |
| SVO | 1.43 | 0.514–3.977 | 0.493 | 0.67 | 0.19–2.398 | 0.538 |
| CE | 0.975 | 0.28–3.392 | 0.968 | 2.12 | 0.433–10.39 | 0.354 |
| Others | 11.05 | 1.307–93.38 | 0.027 | 14.89 | 1.35–164.31 | 0.028 |

OR: Odds Ratio, 95% CI: 95% Confidence Interval, aOR: Adjusted Odds Ratio, SD: Standard Deviation, NIHSS: National Institutes of Health Stroke Scale, IQR: Interquartile Range, TIA: Transient Ischemic Attack, TOAST: Trial of Org 10172 in Acute Stroke Treatment, LAA: Large Artery Atherosclerosis, SVO: Small Vessel Occlusion, CE: Cardioembolism

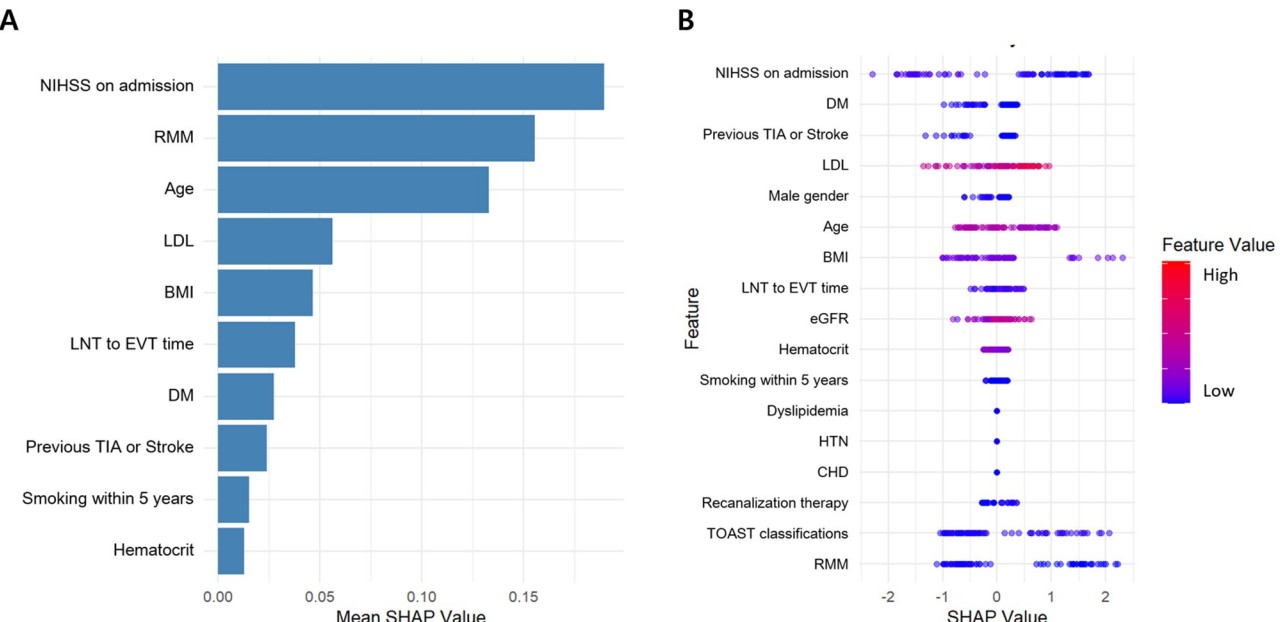

**Fig 1. Variable importance and SHAP summary plot for predicting 3-month functional outcomes in acute ischemic stroke patients.** (A) Variable importance plot based on the mean absolute SHAP values demonstrates the relative contribution of each feature to the model's predictions. NIHSS on admission and RMM were the most influential predictors, followed by age, LDL-cholesterol, BMI, and Onset-to Door time. (B) SHAP summary plot shows the impact of each feature on the predicted outcomes. Each dot represents an individual patient, with the color gradient indicating the feature value (blue = low, red = high). Positive SHAP values indicate a contribution toward unfavorable outcomes, while negative SHAP values suggest a contribution toward favorable outcomes. NIHSS on admission and RMM consistently showed the strongest positive associations with poor outcomes, while other factors, including age, LDL cholesterol, and BMI, exhibited more complex, non-linear relationships. SHAP: Shapley Additive exPlanations, NIHSS: National Institutes of Health Stroke Scale, RMM: Reduced Muscle mass, LDL: Low-Density Lipoprotein, BMI: Body Mass Index, TIA: Transient Ischemic Attack, DM: Diabetes Mellitus, eGFR: estimated Glomerular Filtration Rate.

and poor nutritional status, all of which can adversely affect post-stroke recovery. Additionally, RMM is linked to age, reduced physical activity, and poorer cardiovascular health, all of which may exacerbate neurological deficits and hinder functional recovery. In our study, patients in the RMM group were older and had significantly higher initial stroke severity (as measured by NIHSS) compared to those in the NMM group. This suggests that RMM is not only associated with post-stroke recovery but also closely linked to the severity of neurological deficits at stroke onset, a well-established determinant of stroke outcomes. This observation aligns with a previous study, which reported that patients with prestroke sarcopenia had significantly higher odds (adjusted OR of 3.54) of experiencing moderate-to-severe strokes (NIHSS score > 5) [11]. Several mechanisms may explain the association between sarcopenia and increased stroke severity at onset. Firstly, sarcopenia is characterized by diminished muscle function and strength, which can impair the patient's ability to compensate for neurological deficits resulting from the stroke. Secondly, existing literature suggests that sarcopenia is linked with chronic low-grade inflammation, which not only contributes to muscle mass degradation but may also exacerbate injury to the ischemic penumbra during acute ischemic events, thereby worsening neurological outcomes, potentially worsening neurological outcomes [11, 12]. However, in our study, there was no significant difference in the high-sensitive C-reactive protein levels between the RMM and NMM groups, indicating that inflammation may not be the primary factor contributing to the increased stroke severity observed in our data. Additionally, sarcopenic individuals typically exhibit lower physical activity levels, which is a known risk factor for both the occurrence and severity of stroke.

Reduced physical activity can lead to poor cardiovascular health, increasing the vulnerability of the brain to ischemic injury and resulting in more severe strokes. A recent study further supports this, demonstrating a causal relationship in patients with elderly stroke [13]. To further validate this strong correlation, we conducted an independent variable importance analysis using SHAP values. This analysis quantified the relative contribution of each predictor to unfavorable outcomes. RMM emerged as the second most influential predictor, following the NIHSS score on admission, and demonstrated a substantial impact on unfavorable outcomes. SHAP summary plots and variable importance plots visually confirmed this finding, illustrating both the magnitude and direction of RMM's effect. These results provide robust evidence of RMM's critical role, independent of other covariates, and offer additional validation of its prognostic significance. This is consistent with previous research indicating that the presence of prestroke RMM adversely affects recovery following acute ischemic stroke [6, 7]. The disproportionate influence of RMM compared to other recognized predictors, such as age, BMI, and LDL cholesterol, emphasizes its unique role in predicting unfavorable outcomes. Unlike traditional risk factors, which may have indirect effects, RMM appears to directly impair recovery through multiple pathways, including reduced strength and endurance, impaired rehabilitation potential, and heightened vulnerability to adverse metabolic and cardiovascular states. Interestingly, non-linear relationships observed for variables like BMI and LDL-cholesterol suggest that their effects on stroke recovery may depend on specific thresholds or interactions with other factors. While variables such as Onset-to-Door time and diabetes mellitus showed relatively lower SHAP values, they may still play a context-dependent role in influencing outcomes. These findings reinforce the need for early identification and management of sarcopenia in acute ischemic stroke patients. Interventions targeting muscle mass preservation, such as resistance training and optimized nutritional support, could potentially mitigate the detrimental impact of RMM on functional recovery. Furthermore, the SHAP analysis demonstrates the utility of advanced machine learning approaches for uncovering complex, non-linear relationships between predictors and outcomes, providing valuable insights to guide clinical decision-making and personalized rehabilitation strategies. In our data, RMM group exhibited lower initial blood pressure, particularly diastolic blood pressure (DBP). The reason for the significantly lower blood pressure observed in the RMM group is unclear. While this could be a coincidental finding, it is also possible that it reflects a reduced acute hypertensive response in the early phase of stroke. The acute hypertensive response is defined as an elevation of blood pressure above normal levels within the first 24 hours after stroke onset and is reported in approximately 60% of stroke patients [14]. This response has been associated with stroke severity at presentation and patient outcomes. The response is thought to result from increased sympathetic nervous system activity and renin release, leading to vasoconstriction. Acute stress reactions, cerebral edema, and increased intracranial pressure may also contribute to this hypertensive response [14]. In our study, despite the statistically significantly higher initial NIHSS scores in the RMM group, there was a tendency toward lower blood pressure. Previous studies have reported that vascular calcification and endothelial dysfunction may contribute to the development of sarcopenia. In fact, increased arterial stiffness, indicated by elevated pulse wave velocity, and reduced endothelial function have been observed in sarcopenic patients [15, 16]. These vascular changes might explain the possibility of a diminished acute hypertensive response during the early phase of stroke in this population. During the acute phase of cerebral infarction, maintaining cerebral perfusion by avoiding an excessive drop in blood pressure is crucial. Therefore, relatively lower initial blood pressure could negatively affect the maintenance of cerebral perfusion. However, while an increase in initial blood pressure is often associated with greater stroke severity, it is generally linked to poor prognosis. In our

study, it remains unclear whether the relatively lower blood pressure in RMM patients is related to their poor outcomes. Further research is warranted to clarify this association.

In comparison to previous studies involving Asian populations, our study revealed a notably higher proportion of acute ischemic stroke patients with RMM. Several factors may contribute to this observation. Firstly, demographic differences may play a role. Our cohort consisted primarily of older adults, a group known to have a higher prevalence of sarcopenia. Previous studies demonstrated that age-related declines in skeletal muscle mass are pronounced in the elderly, particularly in individuals above 70 years of age [17, 18]. Given that our patient cohort had a median age in the higher range, this might explain the elevated proportion of low SMI in comparison to studies involving younger populations or those with a broader age range. Secondly, methodological differences in the assessment of muscle mass likely contribute to the variation in results. We utilized DXA (dual-energy X-ray absorptiometry) for the precise quantification of appendicular skeletal muscle mass (ASM), which is considered the gold standard for sarcopenia diagnosis [18]. Other studies may have used alternative methods, such as bioelectrical impedance analysis, which may yield different prevalence rates due to its lower accuracy in muscle mass assessment, particularly in populations with fluctuating hydration levels. Another possible explanation is comorbidity prevalence. A higher incidence of chronic diseases, such as diabetes, hypertension, and cardiovascular diseases, is associated with greater muscle mass reduction in elderly populations [18]. Our cohort included a significant proportion of patients with multiple comorbidities, which may have further contributed to the higher rates of RMM.

In this study, sarcopenia was diagnosed using DXA, a highly precise and reproducible method. DXA and bioelectrical impedance analysis (BIA) are the most commonly used modalities in Asia, with AWGS 2014 cutoffs widely applied in clinical practice. Multi-frequency BIA demonstrates strong correlations with DXA-measured ASM, offering a cost-effective alternative, although home-use BIA devices are not recommended due to limited accuracy [8]. DXA has notable advantages, including simplicity, speed, and reproducibility, making it suitable for clinical applications. However, its inability to assess muscle quality, such as fat infiltration, and its reliance on patient hydration status are limitations [19]. SMI, visceral adipose tissue (VAT), and subcutaneous adipose tissue (SAT) measurements obtained using computed tomography (CT) and magnetic resonance imaging (MRI) at the L3 vertebral level have been shown to be reliable indicators of body composition. CT remains a gold standard for its accuracy in assessing both muscle mass and quality. MRI has been validated as a reliable alternative, particularly for patients requiring radiation-free imaging. MRI measurements of SMI, VAT, and SAT, particularly from T2-weighted images, show high correlation with CT, with slight variations in VAT quantification [20]. The choice of imaging modality should consider the balance between accuracy, accessibility, and clinical utility. While CT and MRI remain invaluable tools in research, broader adoption in clinical practice requires standardization and cost-effectiveness. Future efforts should focus on optimizing the use of these modalities and establishing uniform diagnostic thresholds to enhance sarcopenia management and improve patient outcomes.

Our study has some limitations. First, the sample size was relatively small, which may limit the generalizability of our findings. The study was supported by a grant covering the costs of DXA scans, which limited the number of enrolled patients. A larger sample size and extended study period would likely strengthen the observed correlations. Furthermore, our study primarily included patients who were able to voluntarily provide consent and had sufficient physical performance, thus likely resulting in a cohort with relatively mild symptoms. This selection bias could further restrict the generalizability of our results. Larger, multicenter studies are needed to validate these results and explore the long-term effects of sarcopenia on stroke recovery. Second, while we measured muscle mass using dual-energy X-ray

absorptiometry (DXA), which is considered a gold standard method, the assessment of muscle strength, such as handgrip strength, was challenging in this acute stroke population. Many patients had cognitive impairments or physical limitations that affected the accuracy of strength measurements. Future studies should explore alternative measures of muscle function that can be reliably applied in stroke patients. Third, this study only assessed 3-month functional outcomes without long-term follow-up data, limiting the evaluation of sustained effects of RMM. The small sample size reduced the power to detect significant associations with infrequent outcomes, such as major adverse cardiovascular events (MACE).

In conclusion, this study demonstrates that reduced muscle mass at the onset of stroke is a significant and independent predictor of poor functional outcomes at 3 months. Recognizing RMM at the time of stroke provides a critical opportunity for closer surveillance and selective rehabilitation. However, beyond acute management, early identification of sarcopenia in high-risk populations, such as those with metabolic syndrome, diabetes, or cardiovascular risk factors, could guide interventions like resistance training, nutritional optimization, and physical activity promotion. These measures could help maintain muscle mass and quality, potentially lowering the risk of future strokes or other adverse outcomes. Furthermore, public health strategies targeting sarcopenia through education, community-based exercise programs, and routine screening in older adults could play a role in primordial prevention. By integrating sarcopenia screening into primary care, clinicians can identify individuals at risk before clinical events occur, enabling timely and effective interventions. We hope this addition highlights the broader clinical implications of sarcopenia assessment and its potential to inform both acute care and long-term prevention strategies.

## Supporting information

**S1 Fig. Flow sheet showing the study design and patients exclusion criteria.**
(DOCX)

**S1 Table. Comparison of laboratory findings and clinical characteristics between reduced and normal muscle mass groups in stroke patients.**
(DOCX)

## Acknowledgments

We would like to express our gratitude to all the patients and their families who participated in this study.

## Author Contributions

**Conceptualization:** Seunguk Jung.

**Data curation:** Seunguk Jung, Eun Bin Cho, Tae-Won Yang.

**Formal analysis:** Kyong Young Kim, Seunguk Jung.

**Funding acquisition:** Seunguk Jung.

**Investigation:** Kyong Young Kim, Seunguk Jung.

**Methodology:** Eun Bin Cho, Tae-Won Yang.

**Supervision:** Seunguk Jung.

**Writing – original draft:** Kyong Young Kim.

**Writing – review & editing:** Seunguk Jung, Seung Joo Kim, Hyunsung Kim, Sunhye Jung.

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
