## [Decision Letter · Decision Letter 0]

13 Nov 2024

PONE-D-24-44218The Impact of Reduced Skeletal Muscle Mass at Stroke Onset on 3-Month Functional Outcomes in Acute Ischemic Stroke PatientsPLOS ONE

Dear Dr. Jung,

Thank you for submitting your manuscript to PLOS ONE. After careful consideration, we feel that it has merit but does not fully meet PLOS ONE’s publication criteria as it currently stands. Therefore, we invite you to submit a revised version of the manuscript that addresses the points raised during the review process.

This paper contains interesting content, but it needs to be revised in accordance with the reviewer's comments. Please read the suggestions carefully and respond accordingly.

We look forward to receiving your revised manuscript.

Kind regards,

Masaki Mogi

Academic Editor

PLOS ONE

“This research was supported by the Lee Jung Ja research grant of Gyeongsang National University Hospital (LJJ-GNUH-2019-0001).”

4. Please ensure that you include a title page within your main document. You should list all authors and all affiliations as per our author instructions and clearly indicate the corresponding author.

Reviewers' comments:

Reviewer's Responses to Questions

**Comments to the Author**

1. Is the manuscript technically sound, and do the data support the conclusions?

Reviewer #1: Yes

2. Has the statistical analysis been performed appropriately and rigorously? 

Reviewer #1: Yes

3. Have the authors made all data underlying the findings in their manuscript fully available?

Reviewer #1: Yes

4. Is the manuscript presented in an intelligible fashion and written in standard English?

Reviewer #1: Yes

5. Review Comments to the Author

Reviewer #1: Congratulations on this effort! This work explores an important aspect of stroke rehabilitation and deserves an audience, pending satisfactory responses to the following concerns:

The inclusion criteria need to be more elaborate. "No significant physical impairments" is definitely not sufficient.

While mRS is undoubtedly an important outcome at 90 days, what about associations with other adverse outcomes? Given such a strong correlation with RMM, it's likely that there would be strong correlations with other similar outcomes, e.g., MACE. It would be worth mentioning these.

Please justify the sample size adequacy to prove this association. Given that this is a prospective cohort with what sounds like a high volume of eligible patients, could you have extended the period to generate a stronger correlation with sample size adequacy?

Please discuss how DXA compares with other established markers of sarcopenia, such as L3SMI. DXA introduces an additional investigation to the diagnostic process. Is there a way to satisfactorily diagnose RMM without requiring extra investigations? What are the cost implications of using DXA for this purpose?

While this is undoubtedly an important correlation, please add a section in the discussion regarding how it can change clinical practice. Given that RMM is identified at the time of stroke, there's not much we could do after the event except closer surveillance and selective rehabilitation. Is there anything you think is possible in terms of primary/primordial prevention?

Please address the remarkably strong association with RMM (OR >8) while other recognised markers of poor prognosis barely reach an OR of 1. How do you justify this finding? Are there additional analyses you could perform to further validate this very strong correlation? Perhaps an independent variable importance analysis?

Having said the above, this is indeed a well-executed study with good statistical analysis and definitely explores a strong association with significant potential to identify and treat patients at high risk of debilitating stroke.

6. PLOS authors have the option to publish the peer review history of their article (what does this mean?). If published, this will include your full peer review and any attached files.

Reviewer #1: **Yes: **Bharadhwaj Ravindhran

---

## [Author Response · Author response to Decision Letter 0]

25 Dec 2024

Journal requirements : 

1. Please ensure that your manuscript meets PLOS ONE’s style requirements, including those file naming. 

Response) Thank you for pointing out the need for compliance with PLOS ONE’s style requirements. We have carefully reviewed and revised the manuscript to align with these guidelines. The files have been renamed according to PLOS ONE’s naming conventions. The title page, author affiliations, and main body text have been formatted accordingly.

Response) As requested, we have included the following statement in the manuscript:

Response) Ethical restrictions limit the sharing of de-identified data due to the inclusion of sensitive patient information. These restrictions were imposed by our Institutional Review Board (IRB approval No. 2019-03-005). Requests for data access can be directed to the IRB at https://www.r-bay.co.kr or via telephone at 82-55-214-3701. This has been clarified in the manuscript.

4. Please ensure that you include a title page within your main document. You should list all authors and all affiliations as per our author instructions and clearly indicate the corresponding author.

Response) A title page has been added to the main document, listing all authors and their affiliations. The corresponding author has also been clearly indicated.

Response) Captions for all Supporting Information files have been added at the end of the manuscript, and in-text citations have been updated accordingly.

Responses to the Reviewers’ comments

1. The inclusion criteria need to be more elaborate. "No significant physical impairments" is definitely not sufficient.

Response) Thank you for highlighting the need for more detailed inclusion criteria. The following revision has been made in the manuscript:

"We included patients aged 18 years or older with acute ischemic stroke confirmed by CT or MRI, presenting within 7 days of onset, with premorbid mRS scores below 2, and without conditions significantly affecting skeletal muscle mass (e.g., severe immobility or chronic musculoskeletal disorders)."

The updated content has been bolded in the revised manuscript for clarity.

2. While mRS is undoubtedly an important outcome at 90 days, what about associations with other adverse outcomes? Given such a strong correlation with RMM, it's likely that there would be strong correlations with other similar outcomes, e.g., MACE. It would be worth mentioning these.

Response) We appreciate this comment. While the study primarily focused on 3-month functional outcomes, the occurrence of other major adverse cardiovascular events (MACE) was infrequent due to the small sample size. This limitation has been acknowledged as follows:

"Third, this study only assessed 3-month functional outcomes without long-term follow-up data, limiting the evaluation of sustained effects of reduced muscle mass (RMM). The small sample size reduced the power to detect significant associations with infrequent outcomes, such as major adverse cardiovascular events (MACE)."

Additionally, data on the occurrence of cardiovascular events and cerebrovascular accidents (CVAs) within the first 3 months have been added to the supporting data for further clarification. This addition is now bolded in the manuscript.

3. Please justify the sample size adequacy to prove this association. Given that this is a prospective cohort with what sounds like a high volume of eligible patients, could you have extended the period to generate a stronger correlation with sample size adequacy?

Response) The study’s sample size was limited by the funding allocated for DXA scans. We have added this explanation in the discussion:

"The study was supported by a grant covering the costs of DXA scans, which limited the number of enrolled patients. A larger sample size and extended study period would likely strengthen the observed correlations."

4. Please discuss how DXA compares with other established markers of sarcopenia, such as L3SMI. DXA introduces an additional investigation to the diagnostic process. Is there a way to satisfactorily diagnose RMM without requiring extra investigations? What are the cost implications of using DXA for this purpose?

Response) A detailed comparison of DXA with alternative sarcopenia markers, such as L3SMI and BIA, has been added to the discussion. This comparison addresses the cost implications and feasibility of DXA in clinical settings:

"In this study, sarcopenia was diagnosed using DXA, a highly precise and reproducible method. DXA and bioelectrical impedance analysis (BIA) are the most commonly used modalities in Asia, with AWGS 2014 cutoffs widely applied in clinical practice. Multi-frequency BIA demonstrates strong correlations with DXA-measured ASM, offering a cost-effective alternative, although home-use BIA devices are not recommended due to limited accuracy.(8) DXA has notable advantages, including simplicity, speed, and reproducibility, making it suitable for clinical applications. However, its inability to assess muscle quality, such as fat infiltration, and its reliance on patient hydration status are limitations.(19) SMI, visceral adipose tissue (VAT), and subcutaneous adipose tissue (SAT) measurements obtained using computed tomography (CT) and magnetic resonance imaging (MRI) at the L3 vertebral level have been shown to be reliable indicators of body composition. CT remains a gold standard for its accuracy in assessing both muscle mass and quality. MRI has been validated as a reliable alternative, particularly for patients requiring radiation-free imaging. MRI measurements of SMI, VAT, and SAT, particularly from T2-weighted images, show high correlation with CT, with slight variations in VAT quantification.(20) The choice of imaging modality should consider the balance between accuracy, accessibility, and clinical utility. While CT and MRI remain invaluable tools in research, broader adoption in clinical practice requires standardization and cost-effectiveness. Future efforts should focus on optimizing the use of these modalities and establishing uniform diagnostic thresholds to enhance sarcopenia management and improve patient outcomes.”

5. While this is undoubtedly an important correlation, please add a section in the discussion regarding how it can change clinical practice. Given that RMM is identified at the time of stroke, there's not much we could do after the event except closer surveillance and selective rehabilitation. Is there anything you think is possible in terms of primary/primordial prevention?

Response) Thank you for your insightful feedback on how sarcopenia assessment can influence clinical practice, particularly in the context of primary and primordial prevention. In response to your comment, we have added this in the discussion to address this important aspect. 

“Recognizing RMM at the time of stroke provides a critical opportunity for closer surveillance and selective rehabilitation. However, beyond acute management, early identification of sarcopenia in high-risk populations, such as those with metabolic syndrome, diabetes, or cardiovascular risk factors, could guide interventions like resistance training, nutritional optimization, and physical activity promotion. These measures could help maintain muscle mass and quality, potentially lowering the risk of future strokes or other adverse outcomes. Furthermore, public health strategies targeting sarcopenia through education, community-based exercise programs, and routine screening in older adults could play a role in primordial prevention. By integrating sarcopenia screening into primary care, clinicians can identify individuals at risk before clinical events occur, enabling timely and effective interventions. We hope this addition highlights the broader clinical implications of sarcopenia assessment and its potential to inform both acute care and long-term prevention strategies.”.

6. Please address the remarkably strong association with RMM (OR >8) while other recognised markers of poor prognosis barely reach an OR of 1. How do you justify this finding? Are there additional analyses you could perform to further validate this very strong correlation? Perhaps an independent variable importance analysis?

Response) The remarkably strong association between reduced muscle mass (RMM) and poor outcomes (OR > 8) has been further justified and validated through additional analysis using SHAP (SHapley Additive exPlanations) values. The sensitivity analysis, previously included in the initial manuscript (weighted linear regression models using overlap weights), and Figure 1 from the initial draft were excluded from this version due to statistical limitations and insufficient reliability. Instead, SHAP analysis has been conducted to provide robust evidence of the findings, as detailed below.

SHAP Analysis Findings and new figure (Added to Results Section):

 “Given the strong association of RMM with poor outcomes, this relationship may partly reflect its connection to other prognostic factors. To investigate this further, we conducted an independent variable importance analysis using SHAP (SHapley Additive exPlanations) values. This method quantified RMM's unique contribution to outcome prediction, independent of its correlation with other variables. SHAP analysis identified RMM as the second most influential predictor of unfavorable outcomes, following the NIHSS score on admission. SHAP summary and variable importance plots (Fig. 1) visually confirmed these findings, highlighting RMM's substantial independent impact. The SHAP summary plot (Fig. 1B) revealed that NIHSS on admission contributed most to poor outcomes, with higher scores (red) consistently linked to unfavorable results. RMM, the second-ranked variable, showed positive SHAP values, emphasizing its robust independent association with poor prognosis. Age was the third most significant predictor, with older age contributing to worse outcomes. LDL cholesterol and BMI demonstrated moderate influence with non-linear effects, reflecting the complexity of their roles in recovery. Other factors, such as time from symptom onset to endovascular therapy (LNT to EVT time) and diabetes mellitus (DM), had lower SHAP values, indicating a more limited but context-dependent impact. The variable importance plot (Fig. 1A) reinforced these findings, with NIHSS on admission and RMM showing the highest mean absolute SHAP values.”

Added Discussion Points: 

" The remarkably strong association between reduced muscle mass (RMM) and unfavorable outcomes (OR > 8) compared to other recognized markers, which displayed relatively modest odds ratios, may partly be explained by the close relationship between RMM and other factors influencing poor prognosis including age. Reduced skeletal muscle mass and function is associated with systemic inflammation, metabolic dysfunction, and poor nutritional status, all of which can adversely affect post-stroke recovery. Additionally, RMM is linked to age, reduced physical activity, and poorer cardiovascular health, all of which may exacerbate neurological deficits and hinder functional recovery. In our study, patients in the RMM group were older and had significantly higher initial stroke severity (as measured by NIHSS) compared to those in the NMM group. This suggests that RMM is not only associated with post-stroke recovery but also closely linked to the severity of neurological deficits at stroke onset, a well-established determinant of stroke outcomes."

“To further validate this strong correlation, we conducted an independent variable importance analysis using SHAP values. This analysis quantified the relative contribution of each predictor to unfavorable outcomes. RMM emerged as the second most influential predictor, following the NIHSS score on admission, and demonstrated a substantial impact on unfavorable outcomes. SHAP summary plots and variable importance plots visually confirmed this finding, illustrating both the magnitude and direction of RMM's effect. These results provide robust evidence of RMM's critical role, independent of other covariates, and offer additional validation of its prognostic significance. This is consistent with previous research indicating that the presence of prestroke RMM adversely affects recovery following acute ischemic stroke.(6, 7) The disproportionate influence of RMM compared to other recognized predictors, such as age, BMI, and LDL cholesterol, emphasizes its unique role in predicting unfavorable outcomes. Unlike traditional risk factors, which may have indirect effects, RMM appears to directly impair recovery through multiple pathways, including reduced strength and endurance, impaired rehabilitation potential, and heightened vulnerability to adverse metabolic and cardiovascular states. Interestingly, non-linear relationships observed for variables like BMI and LDL-cholesterol suggest that their effects on stroke recovery may depend on specific thresholds or interactions with other factors. While variables such as time from symptom onset to treatment (LNT to EVT time) and diabetes mellitus showed relatively lower SHAP values, they may still play a context-dependent role in influencing outcomes. These findings reinforce the need for early identification and management of sarcopenia in acute ischemic stroke patients. Interventions targeting muscle mass preservation, such as resistance training and optimized nutritional support, could potentially mitigate the detrimental impact of RMM on functional recovery. Furthermore, the SHAP analysis demonstrates the utility of advanced machine learning approaches for uncovering complex, non-linear relationships between predictors and outcomes, providing valuable insights to guide clinical decision-making and personalized rehabilitation strategies”

Updated Methodology Section: 

"Independent variable importance analysis was conducted using SHAP (SHapley Additive exPlanations) values to

---

## [Decision Letter · Decision Letter 1]

2 Jan 2025

The impact of reduced skeletal muscle mass at stroke onset on 3-month functional outcomes in acute ischemic stroke patients

PONE-D-24-44218R1

Dear Dr. Jung,

We’re pleased to inform you that your manuscript has been judged scientifically suitable for publication and will be formally accepted for publication once it meets all outstanding technical requirements.

Kind regards,

Masaki Mogi

Academic Editor

PLOS ONE

Additional Editor Comments (optional):

Reviewers' comments:

Reviewer's Responses to Questions

**Comments to the Author**

1. If the authors have adequately addressed your comments raised in a previous round of review and you feel that this manuscript is now acceptable for publication, you may indicate that here to bypass the “Comments to the Author” section, enter your conflict of interest statement in the “Confidential to Editor” section, and submit your "Accept" recommendation.

Reviewer #1: All comments have been addressed

2. Is the manuscript technically sound, and do the data support the conclusions?

Reviewer #1: Yes

3. Has the statistical analysis been performed appropriately and rigorously? 

Reviewer #1: Yes

4. Have the authors made all data underlying the findings in their manuscript fully available?

Reviewer #1: Yes

5. Is the manuscript presented in an intelligible fashion and written in standard English?

Reviewer #1: Yes

6. Review Comments to the Author

Reviewer #1: Congratulations on a great effort ! I am happy that my concerns have been addressed satisfactorily !

7. PLOS authors have the option to publish the peer review history of their article (what does this mean?). If published, this will include your full peer review and any attached files.

Reviewer #1: **Yes: **Bharadhwaj Ravindhran

---

## [Editor Report · Acceptance letter]

5 Jan 2025

PONE-D-24-44218R1 

PLOS ONE

Dear Dr. Jung, 

I'm pleased to inform you that your manuscript has been deemed suitable for publication in PLOS ONE. Congratulations! Your manuscript is now being handed over to our production team.

Kind regards, 

on behalf of

Dr. Masaki Mogi 

Academic Editor

PLOS ONE
